# In Vitro Effects of Aminopyridyl Ligands Complexed to Copper(II) on the Physiology and Interaction Process of *Trypanosoma cruzi*

**DOI:** 10.3390/tropicalmed8050288

**Published:** 2023-05-21

**Authors:** Rafaela Silva-Oliveira, Leandro S. Sangenito, Andrew Reddy, Trinidad Velasco-Torrijos, André L. S. Santos, Marta H. Branquinha

**Affiliations:** 1Laboratório de Estudos Avancados de Microrganismos Emergentes e Resistentes (LEAMER), Departamento de Microbiologia Geral, Instituto de Microbiologia Paulo de Goes, Universidade Federal do Rio de Janeiro, Rio de Janeiro 21941-902, Brazil; rafa.oliveira@micro.ufrj.br (R.S.-O.); leandro.sangenito@micro.ufrj.br (L.S.S.); andre@micro.ufrj.br (A.L.S.S.); 2Instituto Federal de Educação, Ciência e Tecnologia do Rio de Janeiro, Nilópolis 26530-060, Brazil; 3Ferrier Research Institute, Victoria University of Wellington, Lower Hutt 5010, New Zealand; andrew.reddy@vuw.ac.nz; 4Department of Chemistry, Maynooth University, W23VP22 Maynooth, Co. Kildare, Ireland; trinidad.velascotorrijos@nuim.ie; 5The Kathleen Lonsdale Institute for Human Health Research, Maynooth University, W23VP22 Maynooth, Co. Kildare, Ireland; 6Programa de Pós-Graduação em Bioquímica, Instituto de Química, Universidade Federal do Rio de Janeiro, Rio de Janeiro 21941-909, Brazil

**Keywords:** *Trypanosoma cruzi*, Chagas disease, metal complexes, chemotherapy

## Abstract

Chagas disease is derived from the infection by the protozoan *Trypanosoma cruzi*. In many countries, benznidazole is the only drug approved for clinical use despite several side effects and the emergence of resistant parasite strains. In this context, our group has previously pointed out that two novel aminopyridine derivatives complexed with Cu^2+^, namely, *cis*-aquadichloro(*N*-[4-(hydroxyphenyl)methyl]-2-pyridinemethamino)copper (**3a**) and its glycosylated ligand *cis*-dichloro (*N*-{[4-(2,3,4,6-tetra-*O*-acetyl-β-D-glucopyranosyloxy)pheny]lmethyl}-2-pyridinemethamino)copper (**3b**), are effective against *T. cruzi* trypomastigote forms. With this result in mind, the present work aimed to investigate the effects of both compounds on trypomastigotes physiology and on the interaction process with host cells. Apart from loss of plasma membrane integrity, an increased generation of reactive oxygen species (ROS) and decreased mitochondrial metabolism were observed. Pretreatment of trypomastigotes with these metallodrugs inhibited the association index with LLC-MK_2_ cells in a typical dose-dependent manner. Both compounds showed low toxicity on mammalian cells (CC_50_ > 100 µM), and the IC_50_ values calculated for intracellular amastigotes were determined as 14.4 µM for **3a** and 27.1 µM for **3b**. This set of results demonstrates the potential of these aminopyridines complexed with Cu^2+^ as promising candidates for further antitrypanosomal drug development.

## 1. Introduction

The causative agent of Chagas disease is the heteroxenic parasite *Trypanosoma cruzi* (Chagas, 1909) (Kinetoplastida, Trypanosomatidae), which is estimated to affect six million people worldwide and cause 30,000 deaths per year [1]. Transmission is classically through triatomine insects, but other forms are also seen, such as blood transfusion, organ transplantation, and ingestion of contaminated food [2]. The lack of specific symptoms in the acute stage of the disease usually allows it to evolve to the chronic phase, which can appear up to 20 years after infection. Around 30% of patients will develop some sort of chronic phase forms, such as chronic Chagas cardiomyopathy (CCC), which is the biggest cause of non-ischemic cardiomyopathy (NICM) in Latin America and can lead to sudden cardiac death (SCD) in most cases [3]. Chagas disease is considered one of the neglected tropical diseases (NTDs) by the World Health Organization [4], which aims to eliminate these infections as a public health problem by 2030 [5]. The impact is even more important because of the current limited therapeutic arsenal.

Only two therapeutic options are currently available to treat Chagas disease: the nitroheterocyclic compounds benznidazole (BZ) and nifurtimox. In many countries, including Brazil, BZ is used as the unique first-line treatment due to the great uncertainty regarding the effectiveness of nifurtimox [6]. The BZ therapy has a cure rate of 80% of patients in the acute phase, but only 20% in the chronic phase [7]. The low efficacy, emergence of new resistant strains, and many side effects make the only treatment option for this illness unsuitable [8,9]. In this regard, new compounds have been evaluated to seek an alternative to treatment for Chagas disease; however, none of the compounds tested in clinical trials so far has surpassed BZ efficacy [10,11,12]. Therefore, these treatment limitations compel the discovery of novel effective and safe molecules to combat this infection.

In this scenario, the use of metal complexes associated with repurposed/novel drugs is a new line of research that is gaining attention, since many of these compounds can be bactericidal, fungicidal, and even present antiparasitic action [13,14,15,16,17,18]. In this sense, a previous study from our group showed that two aminopyridine derivatives complexed with Cu^2+^, namely, *cis*-aquadichloro(*N*-[4-(hydroxyphenyl)methyl]-2-pyridinemethamino)copper (**3a**) and its glycosylated ligand *cis*-dichloro(*N*-{[4-(2,3,4,6-tetra-*O*-acetyl-β-D-glucopyranosyloxy)pheny]lmethyl}-2-pyridinemethamino)copper (**3b**), were effective against the infective form (trypomastigote) of *T. cruzi*, while non-toxic action was reported for RAW macrophages [19]. Based on these data, the present work aimed to evaluate the effects of both promising metal-based compounds on crucial biological processes of *T. cruzi* trypomastigotes, including plasma membrane integrity, mitochondrial physiology, intracellular redox status, and lipid storage. In addition, the action on the parasite-host cell interaction was evaluated in order to check the possible anti-amastigote effect. 

## 2. Materials and Methods

### 2.1. Mammalian Cells and Parasite Cultivation

LLC-MK_2_ epithelial cells were maintained in Dulbecco’s Modified Eagle’s Medium (DMEM) supplemented with 10% fetal bovine serum (FBS) at 37 °C in an atmosphere containing 5% CO_2_. Tissue culture-derived trypomastigotes of *T. cruzi* Y strain were harvested (500× *g*, 5 min) from culture supernatants of 5-day-old infected LLC-MK_2_ cells. The parasites were counted using a Neubauer chamber and resuspended in fresh medium to a final concentration of 10^6^ viable trypomastigotes per mL. The parasite viability was assessed by motility and lack of Trypan blue staining [20].

### 2.2. Aminopyridines-Derived Metallodrugs

*Cis*-aquadichloro(*N*-[4-(hydroxyphenyl)methyl]-2-pyridinemethamino)copper (**3a**) and its glycosylated ligand, *cis*-dichloro(*N*-{[4-(2,3,4,6-tetra-*O*-acetyl-β-D-glucopyranosyloxy)pheny]lmethyl}-2-pyridinemethamino)copper (**3b**) (Figure 1) used in the present work were synthesized in accordance to published methods [19]. The lethal dose for half of the parasitic population (LD_50_) was previously determined by our group for both metallodrugs, and these were, respectively, 1.7 and 1.8 µM after 24 h of incubation [19]. Dimethyl sulfoxide (DMSO), the solvent of both compounds, was used as a control and did not affect the parasite viability in the concentrations used.

### 2.3. Effects on Plasma Membrane Integrity

Trypomastigotes (cell density of 10^6^ parasites/mL) were treated with the ½ × LD_50_, LD_50_, and 2 × LD_50_ doses of both compounds for 24 h, and then incubated for 5 min with propidium iodide (PI) at 1 mg/mL in phosphate-buffered saline (PBS; 150 mM NaCl, 20 mM phosphate buffer, pH 7.2). Cells were then washed in PBS, harvested at 500× *g*/5 min, dispensed into 96-well opaque plates and analyzed in a spectrofluorometer (SpectraMax Gemini XPS, Molecular Devices, San Jose, CA, USA) with excitation and emission wavelengths of, respectively, 540 and 608 nm. Parasites permeabilized with 4% paraformaldehyde were used as non-viable cells (positive PI-staining).

### 2.4. Effects on Mitochondrial Metabolism

MTT [3-(4,5-dimethylthiazol-2-yl)-2,5-diphenyltetrazolium bromide] assay was employed for mitochondrial metabolism testing in sterile 96-well plates [21]. Trypomastigotes (cell density of 10^6^ parasites/mL) were treated or not (control) with the ½ × LD_50_, LD_50_, and 2 × LD_50_ values of **3a** and **3b** for 24 h. The parasites in each system were counted, resuspended to a final concentration of 10^6^ parasites/mL and then MTT solution (5 mg/mL in PBS, 50 μg/well) was added. Plates were then incubated for 3 h in the dark at 37 °C. After centrifugation at 300× *g* for 8 min, the supernatant was removed, the pellet was dissolved in 200 μL of DMSO and absorbance was measured in a microplate reader (SpectraMax spectrofluorometer, Molecular Devices, San Jose, CA, USA) at 490 nm.

### 2.5. Effects on Mitochondrial Transmembrane Potential

The mitochondrial transmembrane potential (Δψm) of control trypomastigote cells and those treated with the ½ × LD_50_, LD_50_, and 2 × LD_50_ doses of both compounds were investigated using the JC-1 fluorochrome, which is a lipophilic cationic mitochondrial vital dye that becomes concentrated in the mitochondrion in response to Δψm. The dye exists as a monomer at low concentrations, where the emission is at 530 nm (green fluorescence), but at higher concentrations it forms J-aggregates after accumulation in the mitochondrion, where the emission is at 590 nm (red fluorescence). Thus, the fluorescence of JC-1 is considered an indicator of an energized mitochondrial state [22]. After 24 h of treatment, cells were harvested, washed in PBS and added to a reaction buffer (pH 7.2) containing 125 mM sucrose, 65 mM KCl, 10 mM HEPES/K^+^, 2 mM inorganic phosphate, 1 mM MgCl_2_ and 500 µM EGTA. To evaluate the Δψm for each experimental condition, treated parasites were standardized at 10⁶ cells per well in 96-well plates, and incubated with the fluorescent dye JC-1 (5 µg/mL) for 40 min, with readings made every minute using a microplate reader. The relative Δψm value was obtained calculating the ratio between the reading at 590 nm and the reading at 530 nm (590:530 ratio)—since mitochondrial de-energization leads to an accumulation of green fluorescence monomers, the decrease in the red/green fluorescence intensity ratio indicates a collapse in the mitochondrial transmembrane potential. A positive control for depolarization of the mitochondrial membrane was also made, which consisted of a system treated with carbonyl cyanide-4-(trifluoromethoxy)phenylhydrazone (FCCP) at 1 µM, a mitochondrial protonophore. To conclude the reaction, FCCP (2 µM) was added in all systems, in order to collapse their ΔΨm [22].

### 2.6. Effects on Reactive Oxygen Species (ROS) Production

After treatment with the ½ × LD_50_, LD_50_, and 2 × LD_50_ values of compounds **3a** and **3b** for 4 and 24 h, the trypomastigotes (10⁶ cells/mL) were resuspended in PBS and incubated with the cell permeable probe dichlorofluorescein (H_2_DCFDA) (40 µg/mL) for 30 min at 25 °C. After incubation, parasites were harvested at 500× *g*/5 min, resuspended in PBS and immediately analyzed in a spectrofluorometer with excitation and emission wavelengths of 504 and 529 nm, respectively. Parasites treated with 1 mM H_2_O_2_ were used as positive controls of ROS production [23].

### 2.7. Effects on Neutral Lipids Storage

Trypomastigotes (10⁶ cells/mL) treated with the ½ × LD_50_, LD_50_, and 2 × LD_50_ values of both compounds for 24 h were incubated with Nile red (5 mg/mL) for 30 min in PBS. Cells were then washed with PBS, harvested at 500× g/5 min, dispensed into 96-well opaque plates and analyzed in a spectrofluorometer with excitation and emission wavelengths of 485 and 538 nm, respectively [23].

### 2.8. Effects on Interaction Process of Trypomastigotes with Host Cells

Trypomastigotes (5 × 10^5^ cells/mL) were treated with non-lethal doses of both compounds (½ × LD_50_ and LD_50_ values) for 1 h. Parasite viability was maintained after this time interval, as determined by checking motility and lack of Trypan blue staining of trypomastigotes counted in a Neubauer chamber. Untreated and treated parasites were washed with PBS, resuspended in fresh DMEM (containing 2% FBS) and placed in a 3 h-interaction with pre-adhered LLC-MK_2_ epithelial cells in 24-well plates at a parasites/cell ratio of 10:1. Non-adhered parasites were then removed by washing with DMEM. Infected epithelial cells were fixed in Bouin solution and stained with Giemsa. The percentage of infected epithelial cells was determined by randomly counting ≥200 cells on each of triplicate coverslips using bright-field microscopy. The association index was obtained by multiplying the percentage of infected epithelial cells by the number of amastigotes per infected cell.

### 2.9. Effects on Epithelial Cell Viability

The effects of compounds **3a** and **3b** on LLC-MK_2_ epithelial cells viability were measured using MTT assay. Twenty-four hours prior to the treatment, epithelial cells (5 × 10⁴) were allowed to adhere in 96-well plates in DMEM supplemented with 10% FBS at 37 °C, in a 5% CO_2_ atmosphere. Non-adherent cells were removed by washes with PBS and the wells refilled with DMEM medium supplemented with 10% FBS. Then, epithelial cells were incubated for 72 h with decreasing concentrations of both compounds (400 to 3.12 µM) diluted at serial concentrations. Subsequently, the culture medium was discharged and the formation of formazan was measured by adding MTT (5 mg/mL in PBS, 50 µg/well) and incubating the wells for 3 h in the dark at 37 °C. After that, plates were centrifuged at 310× *g* for 8 min, the supernatant removed, the pellets dissolved in 200 µL of DMSO and the absorbance measured in a spectrophotometer at 570 nm (Agilent, Santa Clara, CA, USA). The 50% cytotoxicity inhibitory concentration (CC_50_) was determined by linear regression analysis [23]. The selectivity index (SI) was calculated by dividing the CC_50_ value of mammalian cells by the IC_50_ of trypomastigote and amastigote forms. SI values considered satisfactory were those >10 [24].

### 2.10. Effects on Intracellular Amastigotes

LLC-MK_2_ epithelial cells (5 × 10⁴) were allowed to adhere for 24 h in glass coverslips allocated in 24-well plates containing DMEM and 10% FBS. Thereafter, epithelial cells were infected with trypomastigotes in the proportion of 10:1 (parasites/epithelial cell) for 3 h in DMEM supplemented with 2% FBS. The supernatant was then discharged and the plate washed with DMEM to remove free parasites. Infected epithelial cells were then subjected to treatment with both compounds at concentrations capable of maintaining 90% of host cell viability. In this sense, infected epithelial cells were treated for 72 h with compounds **3a** (25 to 3.12 µM) and **3b** (50 to 6.25 µM), in serial dilutions. After this time, coverslips were washed twice with PBS and the systems fixed with Bouin and stained with Giemsa. The percentage of infected epithelial cells and the association index were determined as previously described. The 50% inhibitory concentration (IC_50_) for amastigotes was determined by linear regression analysis [23].

### 2.11. Statistics

All experiments were performed in triplicate on three independent experimental sets. Data were analyzed statistically by means of one-way analysis of variance (ANOVA) using GraphPad Prism software 6.0 (GraphPad Software Inc., La Jolla, CA, USA). *p* values of 0.05 or less were considered statistically significant. Graphs were made in the same program.

## 3. Results

### 3.1. Cell Viability

In order to correlate the morphological alterations promoted by compounds **3a** and **3b** in *T. cruzi* trypomastigotes [19] to cell viability, in the present work both molecules were initially tested across a range of concentrations (½ × LD_50_, LD_50_, and 2 × LD_50_ values) to compare plasma membrane integrity. To this purpose, trypomastigotes pre-treated for 24 h with both compounds were incubated in the presence of PI, which is a widespread red-fluorescent intercalating DNA probe that is not permeant to live cells. A significant and dose-dependent increase in membrane permeability was detected in treated parasites, averaging four times the PI-labeling in non-treated cells at the highest concentration tested (Figure 2). Both compounds showed comparable activity. 

### 3.2. Mitochondrial Activity

The effects of the aminopyridines-derived metallodrugs on both mitochondrial dehydrogenase activities (Figure 3) and mitochondrial membrane potential (Figure 4) were verified in the present work in order to look more closely to the impacts of these compounds in this vital organelle. The treatment of parasite cells with **3a** and **3b** induced a reduction in the mitochondrial dehydrogenase activity in a concentration-dependent manner (Figure 3), as determined by MTT assay. In this sense, the three concentrations tested (½ × LD_50_, LD_50_, and 2 × LD_50_ values) of **3a** caused a significant reduction in the enzymatic activity of mitochondrial dehydrogenases; the highest concentration led to an approximate 60% decay of activity when compared to untreated trypomastigotes (Figure 3). A significant inhibition was also detected with **3b** at LD_50_ and 2 × LD_50_ values, the latter reducing the activity at approximately 50% (Figure 3).

The effect of both compounds on the mitochondrial membrane potential was analyzed in parasite cells previously treated with the same concentrations described above. Incubation with JC-1 showed that cells treated with the three concentrations tested had a significant and dose-dependent reduction of Δψm (Figure 4) when compared with the control (untreated) parasites, indicating a mitochondrial membrane depolarization. For a proper comparison, parasites were also incubated with the classical inhibitor of mitochondrial function, FCCP, a standard protonophore uncoupler that dissipates the mitochondrial electrochemical H^+^ gradient. At 34 min of reaction, FCCP reduced the ΔΨm in 80%, while compounds **3a** and **3b** induced mitochondrial depolarization at comparable activity by around 50% when used at their LD_50_ values and by 63% at their 2 × LD_50_ values (Figure 4). In addition, pre-incubation with FCCP resulted in decreased mitochondrial staining with JC-1 (Figure 4). 

### 3.3. ROS Production

The ability of the metallodrugs **3a** and **3b** to induce oxidative stress was also investigated using the probe H_2_DCFDA. Trypomastigotes exposed to both compounds for only 4 h showed a slight but significant production of ROS only at their 2 × LD_50_ values, the highest concentration used (Figure 5A). However, after 24 h of drug contact the oxidative stress generated rose significantly in a dose-dependent manner (Figure 5B): at the LD_50_ doses, **3a** and **3b** increased ROS production by 47% and 56%, respectively. Cells treated with H_2_O_2_ were used as positive controls and displayed high fluorescence levels (Figure 5).

### 3.4. Neutral Lipid Accumulation

Using fluorometric quantification after Nile red staining, the augment of neutral lipid content was only verified after treatment of both compounds with the 2 × LD_50_ doses: 21% for **3a** and 37% for **3b** (Figure 6). 

### 3.5. Interaction of Trypomastigotes with LLC-MK_2_ Epithelial Cells

In this set of experiments, trypomastigotes were pre-treated for 1 h with the metallodrugs **3a** and **3b** at the ½ × LD_50_ and LD_50_ values before interaction with host cells. Under these conditions, compound **3a** at the ½ × LD_50_ value was capable of reducing the association index of trypomastigotes with LLC-MK_2_ cells by 23%. Similarly, compound **3b** at the ½ × LD_50_ value was capable of inhibiting the interaction with host cells by 29% (Figure 7). Both compounds showed a clear dose-dependent inhibition profile: the inhibition increased to 65% for **3a** and to 51% to **3b** as the concentration of both compounds was increased to the LD_50_ value (Figure 7). These inhibition profiles were not caused by a decrease in trypomastigote viability, as judged by morphology and motility. 

### 3.6. Survival of Intracellular Amastigotes

Initially, the cytotoxicity of compounds **3a** and **3b** to LLC-MK_2_ cells was assessed by MTT after 72 h of treatment. The CC_50_ values calculated for both metallodrugs were 125.7 µM and 175.2 µM, respectively (Figure 8). Based on the calculated CC_50_ values for LLC-MK_2_ cells and the LD_50_ values for trypomastigote forms, the selectivity index (SI) determined for compound **3a** was 73.9 and for compound **3b,** 97.3. In 72 h-treated LLC-MK_2_ cultures, a significant deleterious effect was observed with **3a** only at concentrations >25 μM, while **3b** presented these effects at >50 μM (Figure 8).

Our results showed that these metallodrugs were also able to reduce the survival of intracellular amastigotes. In these experiments, LLC-MK_2_ infected cells were treated once with compounds **3a** (25 to 3.12 µM) and **3b** (50 to 6.25 µM) and then intracellular amastigotes were quantified after 72 h (Figure 9). Both compounds at their highest doses reduced the association indexes by 66% and 76%, respectively, when compared to their respective untreated systems (Figure 9). A clear dose-dependent effect of both compounds on the infection rate was observed, for which the IC_50_ value for **3a** was calculated as 14.4 μM, and that for **3b** as 27.1 μM. Based on these IC_50_ values, SI values were calculated for amastigote forms: 8.73 for compound **3a** and 6.46 for **3b**. 

## 4. Discussion

In the present work, two aminopyridine derivatives complexed with copper were assayed in *T. cruzi* trypomastigotes using a set of experiments: (i) analysis of cell viability; (ii) determination of mitochondrial functioning and ROS production; (iii) neutral lipid accumulation; (iv) inhibition of trypomastigote invasion in LLC-MK_2_ cells; (v) cytotoxicity to mammalian cells; and (vi) verification of the effects on intracellular amastigotes. The anti-*T. cruzi* activity of the Cu(II) complex **3a** and its glycosylated ligand **3b** has been previously evaluated, showing significantly better efficacy toward trypomastigote forms (LD_50_ values of 1.7 and 1.8 μM, respectively) than the current clinical drug benznidazole (LD_50_ value 3.8 μM) [19]. The treatment of trypomastigotes with both compounds caused relevant morphological changes when compared to the typical appearance of non-treated parasites, including rounding in shape with reduced cell size, swelling of the cell body and shortening or loss of flagellum [19]. The significant increase in PI-labeling of cells after treatment with both compounds, as detected in the present work, indicates that these metallodrugs induced loss of parasite viability in a typical concentration-dependent manner, corroborating the morphological alterations previously observed by our group [19].

The alterations in mitochondrial dehydrogenase activities and ΔΨ*m* loss findings in addition to ROS production, support the notion that the imbalance in mitochondrial functionality can strongly contribute to protozoal death induced by the metallodrugs **3a** and **3b**. A special characteristic of trypanosomatids is the presence of a single elongated mitochondrion in the cytoplasm; therefore, the proper function of this organelle, whose major role is energy production, is fundamental for survival [25]. Experimental studies about the mechanisms of action of conventional and novel antitrypanosomatid compounds point out that mitochondrion can be considered one of the most important targets for drug inhibitory action [25,26]. In response to changes in the intracellular environment by different stress signals, such as the presence of different molecules, hypoxia, oxidative stress, and DNA damage, mitochondria become producers of excessive ROS [27]. The mitochondrial damage observed in the results described herein relates to the fundamental role of this organelle in the overall metabolism of the parasite, which may lead to cell death. Neutral lipid accumulation was also tested in the present work, but only observed in cells treated with the highest concentration tested (2 × LD_50_ doses). So, this may be considered a secondary effect of both compounds.

Mammalian host cell recognition of *T. cruzi* trypomastigotes involves the interaction of numerous parasite and host cell membrane molecules and domains, and is a fundamental step for *T. cruzi* infection development [28]. Once the establishment of stable interactions of trypomastigotes with the vertebrate host relies on parasite’s surface molecules, the reduced association indexes described herein suggest a possible interference of both compounds in the early steps of LLC-MK_2_ cells’ infection, either through blockage of ligand/receptor binding or through suppression of ligand/receptor expression [28]. Our work continued through the determination of the CC_50_ values for both metallodrugs in LLC-MK_2_ cells: 125.7 µM for compound **3a** and 175.2 µM for compound **3b**. In comparison, these compounds exhibited low toxicity toward RAW macrophages as well, with CC_50_ values of 106.2 µM and 127.6 µM, respectively [19]. Based on the calculated CC_50_ values for LLC-MK_2_ cells and the LD_50_ values for trypomastigote forms, attractive selectivity indexes (SI) were determined for compounds **3a** (73.9) and **3b** (97.3).

The promising efficacy of both compounds, with higher toxicity to trypomastigotes compared with mammalian cells, stimulated the study of their effects on the replicative amastigote forms of the parasite. Our results showed that these metallodrugs were able to reduce the survival of intracellular amastigotes, but not at levels considered promising. The IC_50_ value for **3a** was calculated as 14.4 μM, and that for **3b** as 27.1 μM. Based on these IC_50_ values, both compounds showed a higher toxicity to *T. cruzi* amastigotes compared to LLC-MK_2_ cells, resulting in moderately satisfactory SI values: 8.73 for compound **3a** and 6.46 for **3b**. In general, only molecules with SI >10 are currently selected for in vivo studies [24]. Since amastigotes are present in an intracellular protective environment and therefore are less exposed to direct contact with these compounds, this may explain the IC_50_ values calculated for this intracellular form being higher than those calculated for trypomastigotes. Altogether, these data point to the activity of these metallodrugs against the clinically relevant stages of *T. cruzi*, with emphasis on trypomastigote forms.

Several authors have pointed to copper complexes as a promising alternative to fight tropical parasitic infections, such as Chagas disease [14,16,17,29,30,31,32,33,34]. Many are the ligands derived from organic compounds that have been used in coordination chemistry, and the antiparasitic activity is often considerably improved by the presence of the metal ion [18,34]. The combination of metal ions with drugs and organic ligands can enhance their activity while, at the same time, reduce the toxicity and undesired side effects that metal ions on their own could impart [18,19,35]. In this regard, the non-metalated ligands of compounds **3a** and **3b** showed significantly higher LD_50_ values (4.6 and 5.3 μM, respectively) than the corresponding copper complexes (1.7 and 1.8 μM, respectively) [19]. It is also clear that copper, per se, is not responsible for the studied anti-*T. cruzi* activity, since the simple salt CuCl_2_ did not show significant toxicity to the parasite [19].

Although it seems clear that Cu(II) coordination enhances the antiparasitic activity of these ligands, the moderate improvement detected for compounds **3a** and **3b** suggests that metal complexation does not prompt the synergistic effects as reported for some antitrypanosomal metal complexes [16,17,18,30]. In this case, it is proposed that Cu(II) provides additional affinity for the target binding sites [19]. Copper-containing coordination compounds were found to be promising antitumor therapeutic agents that act by various mechanisms, such as inhibition of proteasome activity, telomerase activity, ROS formation, DNA degradation, DNA intercalation, paraptosis, among others [34]. Some of the antitrypanosomal Cu(II) complexes reported in the literature bind to DNA [16,17,18,31]. However, the active compounds **3a** and **3b** were also screened previously for toxicity against larvae of *Galleria mellonella*, a model organism used as a surrogate to probe the response of the human innate immune system, and all the larvae treated with both compounds at concentrations as high as 100 µM kept their viability up to 48 h [19]. This result, together with the high SI values found for trypomastigotes, suggests that both compounds are not likely to affect high affinity DNA binding. In the present work, copper complexes induced significant alterations in the mitochondrial metabolism and ROS production, which may imply this organelle as one of the major targets of both compounds, as detected in many other antitrypanosomal molecules [25,26]. The fact that mammalian cells present multiple mitochondria allows a mechanism of compensation in the presence of altered organelles, which raises specificity [27].

The influence of the addition of a glucosyl moiety to **3a**, originating in **3b**, can be discussed. Although carbohydrate conjugation is an attractive means of improving the selectivity of bioactive molecules, the water-soluble character of saccharides can compromise their ability to cross cellular membranes. Therefore, the use of an acetylated sugar, as it occurs in compound **3b**, with increased hydrophobicity and for which the acetyl groups would undergo hydrolysis after administration, is often preferred [19,36]. In this sense, compounds **3a** and **3b** showed a similar range of antitrypanosomal activity, as determined previously by the LD_50_ values calculated for trypomastigote forms [19] and in the present work by the equivalent action on the loss of parasite viability, mitochondrial metabolism, and interaction of trypomastigotes with LLC-MK_2_ cells. However, compound **3b**, bearing a glycosylated ligand of **3a**, was found to be less toxic to RAW macrophages [19] and LLC-MK_2_ cells than its counterpart **3a.** Since survival of intracellular amastigotes, on the other hand, was higher in the presence of **3b** in comparison to **3a,** this may confirm the trend observed for other classes of antiparasitic compounds where the increase in lipophilicity seems to favor the compound distribution to the intracellular target [37].

## 5. Conclusions

In general, the major disadvantage of the compounds **3a** and **3b** tested herein is their high IC_50_ values for amastigotes and, therefore, their relatively low therapeutic selectivity. A strategy used to decrease toxicity and increase the activity is through structural modifications; consequently, a series of derivatives generated through structure-activity relationship (SAR) studies need to be tested against both extracellular trypomastigotes and intracellular amastigotes in the future. Our results indicate that such compounds can be a useful platform for the design of new antichagasic drugs, since structural modifications may contribute to the improvement of the activity and safety of these molecules.

## Figures and Tables

**Figure 1 tropicalmed-08-00288-f001:**
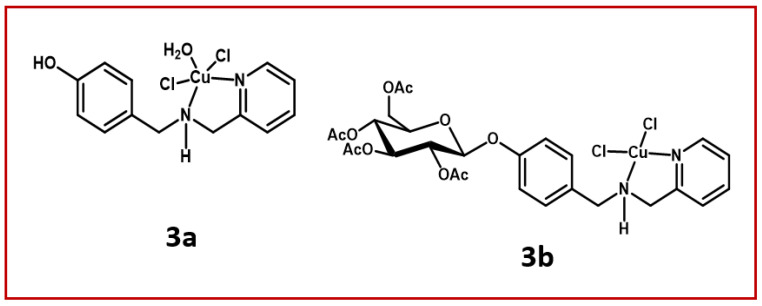
Chemical structures of the metal-based aminopyridines used in the present work: *cis*-aquadichloro(*N*-[4-(hydroxyphenyl)methyl]-2-pyridinemethamino)copper (**3a**) and *cis*-dichloro(*N*-{[4-(2,3,4,6-tetra-*O*-acetyl-β-D-glucopyranosyloxy)pheny]lmethyl}-2-pyridinemethamino)copper (**3b**).

**Figure 2 tropicalmed-08-00288-f002:**
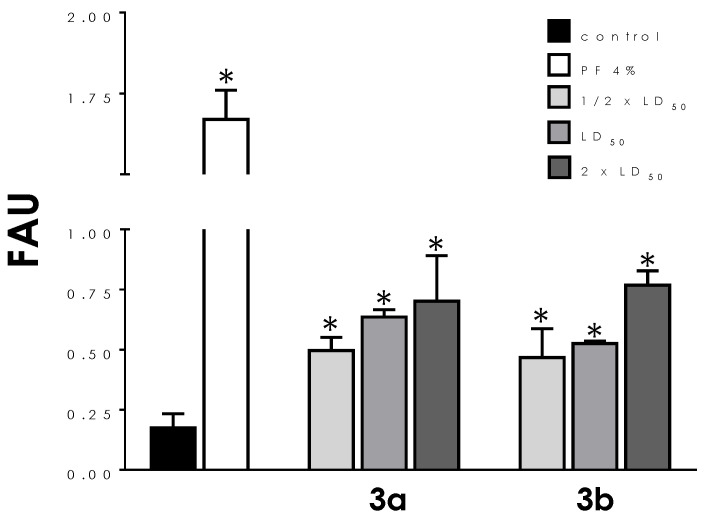
Effects of aminopyridines-derived metallodrugs on the plasma membrane integrity of *T. cruzi* trypomastigotes. The incorporation of propidium iodide (PI) was quantified in a spectrofluorometer in untreated cells (control) and those treated with the metallodrugs **3a** and **3b** in concentrations corresponding to ½ × LD_50_, LD_50_, and 2 × LD_50_ values for 24 h. Results are expressed as fluorescence arbitrary units (FAU). For positive control of non-viable cells (PI-positive staining), parasites were fixed with 4% paraformaldehyde (PF). Values represent mean ± standard deviation of three independent experiments. All treatments showed significant statistical difference (*) compared to control cells (*p* < 0.01).

**Figure 3 tropicalmed-08-00288-f003:**
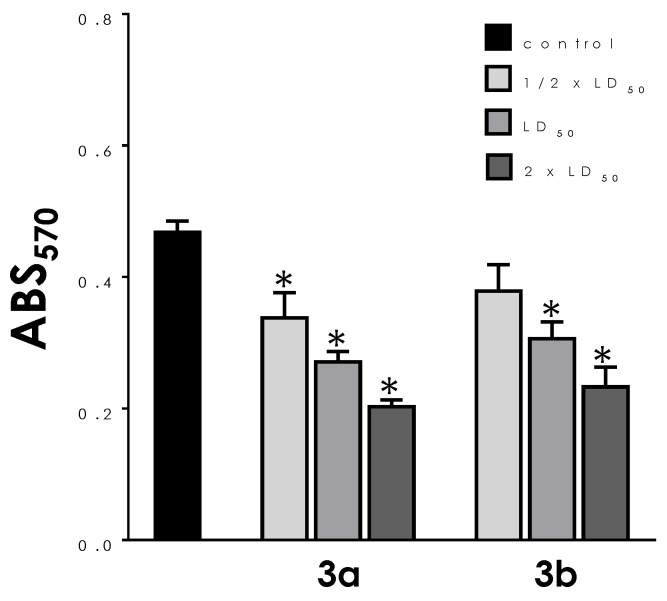
Effects of aminopyridines-derived metallodrugs on the mitochondrial metabolism of *T. cruzi* trypomastigotes. Cells were treated or not (control) with the metallodrugs **3a** and **3b** at concentrations corresponding to ½ × LD_50_, LD_50_, and 2 × LD_50_ values for 24 h. The mitochondrial dehydrogenase activity was determined spectrophotometrically (ABS, absorbance) by MTT assay. Values represent mean ± standard deviation of three independent experiments. The asterisks represent significant statistical differences compared to the control (*p* < 0.01).

**Figure 4 tropicalmed-08-00288-f004:**
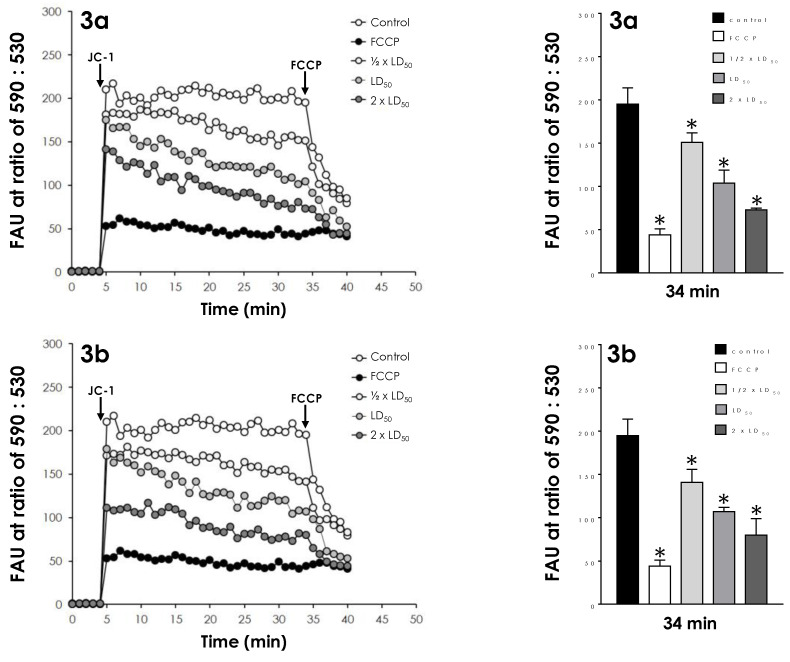
Effects of aminopyridines-derived metallodrugs on ΔΨm in *T. cruzi* trypomastigotes. **Left panels:** After treatment of parasites with the metallodrugs **3a** and **3b** at concentrations corresponding to ½ × LD_50_, LD_50_, and 2 × LD_50_ values for 24 h, the ΔΨ*m* analysis of control and treated parasites was performed using the JC-1 fluorochrome for 30 min. Then, the uncoupler FCCP (2 μM) was added in all systems to collapse mitochondrial potential. The arrows indicate the time points at which JC-1 and FCCP were added. As a positive control of depolarization of the mitochondrial membrane, the reaction was also evaluated in the presence of FCCP (1 μM) throughout the experiment. **Right panels:** Comparison of the ΔΨ*m* values shown in the left panels at 34 min, immediately before the addition of the uncoupler FCCP. The results are expressed as fluorescence arbitrary units (FAU). Values represent mean ± standard deviation of three independent experiments. The asterisks represent significant statistical differences compared to the respective controls (*p* < 0.01).

**Figure 5 tropicalmed-08-00288-f005:**
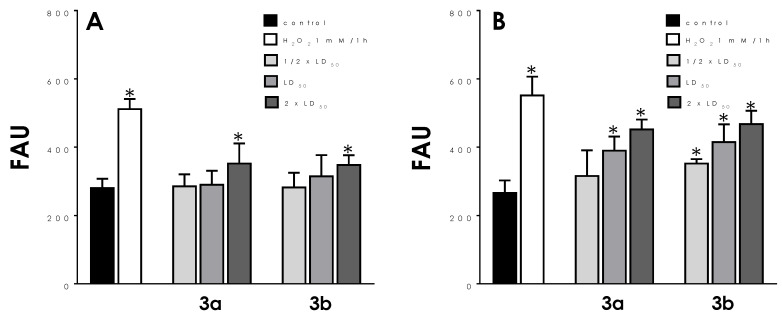
Effects of aminopyridines-derived metallodrugs on ROS production in *T. cruzi* trypomastigotes. The production of ROS was measured fluorometrically in control cells and those treated with the ½ × LD_50_, LD_50_, and 2 × LD_50_ values of the metallodrugs **3a** and **3b** for 4 h (**A**) and 24 h (**B**), using the green fluorescent probe H_2_DCFDA. Cells treated with 1 mM H_2_O_2_ were used as positive control to intracellular generation of ROS. The results are expressed as fluorescence arbitrary units (FAU). Values represent mean ± standard deviation of three independent experiments. The asterisks represent significant statistical differences compared to the respective controls (*p* < 0.01).

**Figure 6 tropicalmed-08-00288-f006:**
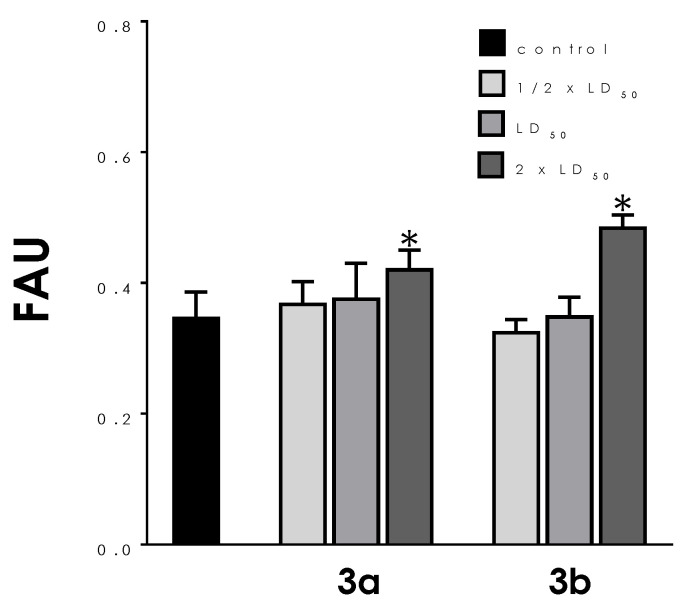
Effects of aminopyridines-derived metallodrugs on lipid accumulation in *T. cruzi* trypomastigotes. The incorporation of Nile red was quantified fluorometrically in the absence (control) or in the presence of the metallodrugs **3a** and **3b** at the ½ × LD_50_, LD_50_, and 2 × LD_50_ values for 24 h. The results are expressed as fluorescence arbitrary units (FAU). Values represent mean ± standard deviation of three independent experiments. The asterisks represent significant statistical differences compared to the respective control (*p* < 0.01).

**Figure 7 tropicalmed-08-00288-f007:**
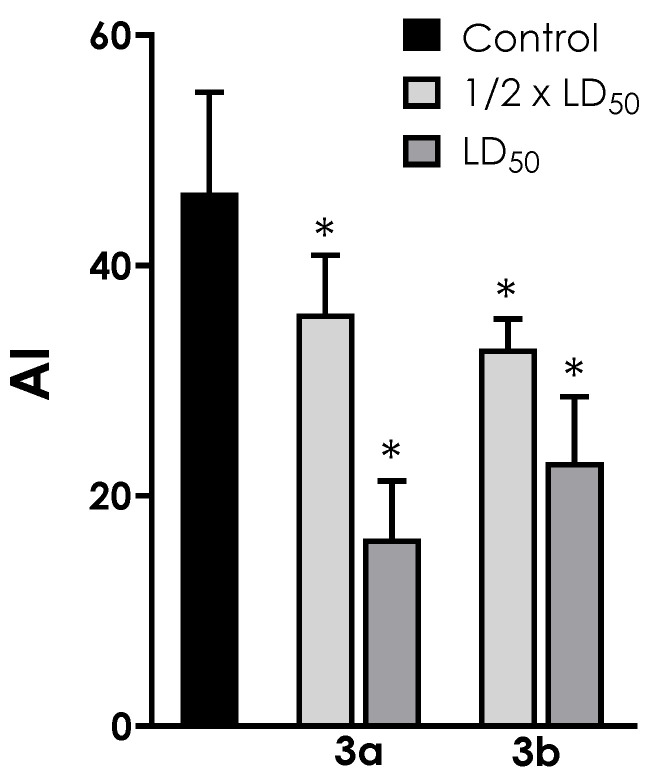
Effects of aminopyridines-derived metallodrugs on the interaction process between *T. cruzi* trypomastigotes and LLC-MK_2_ host cells. Trypomastigotes (5 × 10^5^/^mL^) were treated or not (control) with the metallodrugs **3a** and **3b** at ½ × LD_50_ and LD_50_ doses for 1 h, then washed and incubated with pre-adhered LLC-MK_2_ cells (5 × 10^4^ per well) for 3 h at a parasites/cell ratio of 10:1. The systems were then fixed, stained with Giemsa and counted in an optical microscope. The association index (AI) was determined by multiplying the percentage of infected mammalian cells by the mean number of parasites per infected cell. The asterisks represent significant statistical differences compared to the respective controls (*p* < 0.01).

**Figure 8 tropicalmed-08-00288-f008:**
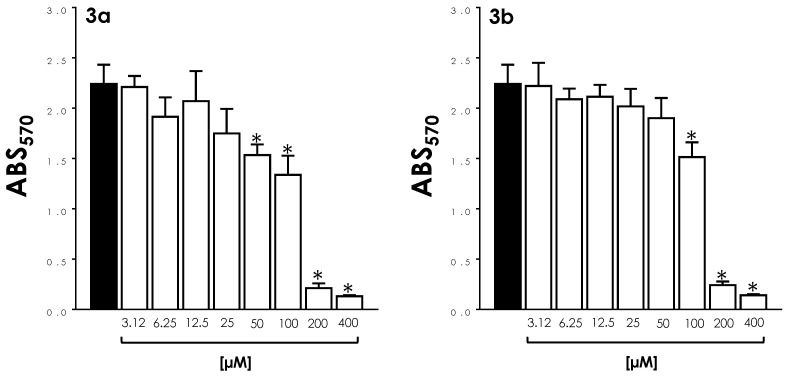
Cytotoxicity of aminopyridines-derived metallodrugs to LLC-MK_2_ epithelial cells. Mammalian cells were incubated for 72 h in the absence (black bars) or in the presence of the metallodrugs **3a** and **3b** at different concentrations (as indicated) (white bars). The viability was then determined spectrophotometrically by measuring the absorbance (ABS) by MTT assay. Data shown are the mean ± standard deviation of three independent experiments performed in triplicate. The asterisks represent significant statistical differences compared to the respective controls (*p* < 0.01).

**Figure 9 tropicalmed-08-00288-f009:**
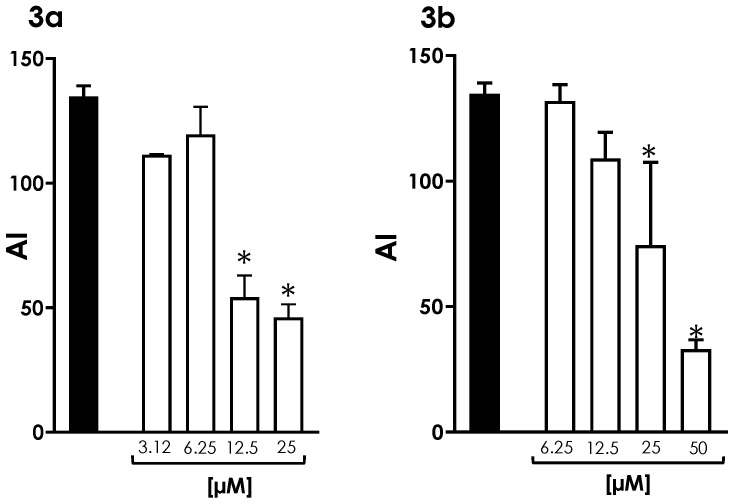
Effects of aminopyridines-derived metallodrugs on *T. cruzi* intracellular amastigotes. LLC-MK_2_ cells were infected with trypomastigotes at a parasites/cell ratio of 10:1 for 3 h at 37 °C. After this period, each system was washed to remove non-internalized parasites and then treated or not (control) with the metallodrugs **3a** and **3b** at different concentrations, as described in the figure, for 72 h. The association index (AI) was determined by multiplying the percentage of infected mammalian cells by the mean number of parasites per infected cell. Data shown are the mean ± standard deviation of three independent experiments performed in triplicate. The asterisks represent significant statistical differences compared to the respective control (*p* < 0.01).

## Data Availability

Not applicable.

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
