# Peer review of "In Vitro Effects of Aminopyridyl Ligands Complexed to Copper(II) on the Physiology and Interaction Process of Trypanosoma cruzi"

_tropicalmed, 2023, doi:10.3390/tropicalmed8050288_

Round 1
Reviewer 1 Report
Branquinha and collaborators describe the activity of two copper(II) complexes against the protozoan T. cruzi, trypomastigota form. The complexes have already been studied previously, by the group, against the amastigote form. The results are expressive with a search for the effective mode of action of these compounds.
I have some suggestions/questions:
1- Size of these compounds, the one with glucosyl moiety is much larger and this may be the factor that leads to lower activity. Was this considered? Is there any evidence of this experimentally?
2- The experiments were conducted in DMSO. Current drugs are administered orally. The inclusion of glucosyl moitey appears to make complex 3b less toxic to RAW macrophages and LLC-MK2, but less water soluble. On page 12 line 42 the authors comment on hydrolysis and cite a reference. Thus, it would be interesting to know if the complexes remain intact in DMSO solution for 48 hours, as well as in the presence of DMEM. Is there interaction with the DMEM medium? If there is hydrolysis, what is the authors' proposal considering this possibility?
3- Page 5, line 18. Item 2.10. Effects on Intracellular Amastigotes. Is it amastigotes or trypomastigota?
4- Figure 5 is not in the manuscript, only the caption.
5- Question: do the complexes generate an imbalance in the mitochondria and, as a result, mitochondrial dysfunctionalization appears? What comes first? Is the stress of the complex and then the mitochondrial dysfunctionalization or the opposite? Do the drugs used BZ and nifurtimox act similarly? How to prove this hypothesis?
Author Response
Branquinha and collaborators describe the activity of two copper(II) complexes against the protozoan T. cruzi, trypomastigota form. The complexes have already been studied previously, by the group, against the amastigote form. The results are expressive with a search for the effective mode of action of these compounds.
I have some suggestions/questions:
1- Size of these compounds, the one with glucosyl moiety is much larger and this may be the factor that leads to lower activity. Was this considered? Is there any evidence of this experimentally?
Answer: We thank the reviewer for the encouraging feedback. As pointed out in the last paragraph of topic 3 (topic 4 in the revised manuscript), the addition of the glucosyl moiety was performed in order to enhance selectivity of the molecule to parasite cells. In this sense, higher CC50 values for LLC-MK2 cells were found for the glycosylated compound 3b, while both molecules displayed similar LD50 values for trypomastigote forms. We may conclude that both compounds can invade parasitic cells in a similar rate, which did not happen in the host cells. These results are the only experimental evidence of the distinct capability of these molecules to invade host cells.
2- The experiments were conducted in DMSO. Current drugs are administered orally. The inclusion of glucosyl moitey appears to make complex 3b less toxic to RAW macrophages and LLC-MK2, but less water soluble. On page 12 line 42 the authors comment on hydrolysis and cite a reference. Thus, it would be interesting to know if the complexes remain intact in DMSO solution for 48 hours, as well as in the presence of DMEM. Is there interaction with the DMEM medium? If there is hydrolysis, what is the authors' proposal considering this possibility?
Answer: Compound 3a is very stable in DMSO/water, as cited in the original manuscript in which the synthesis of these compounds was described (reference 19). As a matter of fact, the glucosyl moiety enhances water solubility, which is counterbalanced by the acetylation – it is the acetyl group that is susceptible to hydrolysis. This sentence was rewritten in the revised version of the manuscript (page 12, lines 551-553). In our experiments, stock solutions were prepared in DMSO, and remained stable for at least 6 months, since no appreciable change in activity was observed, even in DMEM.
3- Page 5, line 18. Item 2.10. Effects on Intracellular Amastigotes. Is it amastigotes or trypomastigota?
Answer: Since host cells were previously infected with trypomastigotes and then both compounds were added to the systems, the effect was determined on amastigote cells.
4- Figure 5 is not in the manuscript, only the caption.
Answer: we thank the reviewer for this observation and apologize for this mistake. Figure 5 was included.
5- Question: do the complexes generate an imbalance in the mitochondria and, as a result, mitochondrial dysfunctionalization appears? What comes first? Is the stress of the complex and then the mitochondrial dysfunctionalization or the opposite? Do the drugs used BZ and nifurtimox act similarly? How to prove this hypothesis?
Answer: It is very difficult to determine the sequence of events occuring in cells when subjected to toxic compounds, especially when pleiotropic effects were detected in microbial cells. In tryponosomatids, the alterations induced in the mitochondrion are usually searched since there is only one of this organelle and the importance of its function in the trypanosomatid metabolism ascertains that it is a vital target in the parasite. One of the possibilities to evatuate the sequence of events should be the use of a concentration gradient of the molecules, which we tried to do. Nifurtimox and Benznidazole are chemically related nitrofurans. Though their mechanism of action is not completely understood, it is suggested that both compounds and their metabolites bind to and block the parasites' antioxidant availability and generate DNA-toxic glyoxal adducts, causing oxidative damage to the parasite (Pérez-Molina et al., 2021, Enferm Infecc Microbiol Clin, 39:458-470). So, at least part of the mechanism of action is shared.
Reviewer 2 Report
The proposed manuscript deals with in vitro assays of two aminopyridine derivatives complexed with copper on trypomastigotes and amastigotes of T. cruzi. The authors refer that, given the results obtained, changes in the chemical structure of the two derivatives can improve the activity and safety of these two molecules. Given the need to develop new drugs to treat Chagas disease, since currently only nifurtimox and benznidazole are available and even then not effective for all chagasic patients, it is necessary to encourage the development of drugs for this disease. It should also be considered that in the chronic phase of Chagas' disease these two drugs have very low efficiency, therefore studies of this nature are relevant and should be supported.
I suggest a small change:
2- Line 38: parasite Trypanosoma cruzi,
I suggest: parasite Trypanosoma cruzi (Chagas, 1909) (Kinetoplastida, Trypanosomatidae).
I also suggest that changes be made to the configurations of figures 2 to 9, in order to facilitate the interpretation, that is, not to use the variations in the shade of the black color, which makes visualization difficult.
Author Response
The proposed manuscript deals with in vitro assays of two aminopyridine derivatives complexed with copper on trypomastigotes and amastigotes of T. cruzi. The authors refer that, given the results obtained, changes in the chemical structure of the two derivatives can improve the activity and safety of these two molecules. Given the need to develop new drugs to treat Chagas disease, since currently only nifurtimox and benznidazole are available and even then not effective for all chagasic patients, it is necessary to encourage the development of drugs for this disease. It should also be considered that in the chronic phase of Chagas' disease these two drugs have very low efficiency, therefore studies of this nature are relevant and should be supported.
I suggest a small change:
2- Line 38: parasite Trypanosoma cruzi, I suggest: parasite Trypanosoma cruzi (Chagas, 1909) (Kinetoplastida, Trypanosomatidae).
Answer: We thank the reviewer for the positive and encouraging feedback. The suggestion was included in the revised manuscript.
I also suggest that changes be made to the configurations of figures 2 to 9, in order to facilitate the interpretation, that is, not to use the variations in the shade of the black color, which makes visualization difficult.
Answer: As a matter of fact, the variations in grey shade were used in order to facilitate visualization. In figures 2-7, the same shade of gray was used for each concentration of both compounds, as well as black bars were used for control cells and white bars for positive controls. So, the comparison of results in each figure is clear. If the shades of gray are removed, comparison of results would be hampered and it would demand the addition of more information in the “x” axis. In Figure 8, each concentration tested on LLC-MK2 cells was represented as a white bar; to standardise this point, all bars in figure 9 were changed to white.
Reviewer 3 Report
Dear Authors,
Thanks a lot for this huge, clear and exhaustive work. I want this work to be published with some minor moves, and especially with this point 3 mixing results and discussion. Your paper will be better understood with point 3 : results and point 4 discussion.
Therefore, It would be very interesting to add clinical informations found in litterature about those therapeutics, in real patients, in vivo. According to me, this has to be add on a specific discussion chapter.
thank you for this work.
Best regards,
Author Response
Dear Authors,
Thanks a lot for this huge, clear and exhaustive work. I want this work to be published with some minor moves, and especially with this point 3 mixing results and discussion. Your paper will be better understood with point 3: results and point 4 discussion.
Answer: We thank the reviewer for the encouraging feedback. As suggested, Results and Discussion are separate topics in the revised version of the manuscript.
Therefore, It would be very interesting to add clinical informations found in litterature about those therapeutics, in real patients, in vivo. According to me, this has to be add on a specific discussion chapter.
Answer: There is no clinical information concerning the use of compounds 3a and 3b. As a matter of fact, this is a future step in our laboratory research.
thank you for this work.
Best regards,